# NumMolFormer: Enhancing Transformer Numerical Reasoning for Functional-Group-Based Molecule Generation

## Abstract

Structure-based drug design critically depends on effectively identifying the active molecular structures. Functional groups serve as local active centers and must be optimally balanced, with excess diminishing specificity and scarcity limiting activity. However, most existing methods model molecules at the atom–bond level rather than at the functional group level, making it difficult to control the quantity of functional groups. To address this, we propose NumMolFormer, a novel molecular generation method that integrates functional group knowledge with numerical modeling. NumMolFormer employs a dual-sequence representation that jointly encodes text sequence tokens of functional groups with their quantitative information, enhanced by a numerical embedding module that leverages symbol–magnitude decomposition and soft magnitude quantization to capture numerical features. Furthermore, we introduce a dual-stream differential attention mechanism to explicitly disentangle textual and numerical contributions. To overcome data scarcity, we build an 18 million molecule dataset with functional group annotations for pretraining, followed by self-supervised and reinforcement learning fine-tuning on protein pockets. Experimental results demonstrate that NumMolFormer can effectively control functional groups in molecular generation and produce molecules with enhanced activity, synthesizability, and drug-likeness when conditioned on protein pockets. The code is available at https://github.com/alan-tsang/NumMolFormer.

## 1 Introduction

Structure-based drug design (SBDD) is an important and widely used approach in AI-assisted drug design (Kim et al., 2020), utilizing protein structures to design molecules with high affinity and favorable drug-like properties (Van Montfort & Workman, 2017; Ferreira et al., 2015). Recent machine learning approaches based on SBDD follow two strategies: sequence-based methods map protein sequences to molecular representations like SMILES (Zhou et al., 2023; Born & Manica, 2023; Grechishnikova, 2021), while graph-based methods model proteins and molecules as graphs using GNNs or diffusion frameworks (Liu et al., 2022b; Guan et al., 2023b; Peng et al., 2022). Despite progress in drug generation, most methods model molecules at the atom-bond level, ignoring medicinal chemistry knowledge. Molecular function is often determined by local active structures—functional groups, rather than individual atoms (Lin & Lu, 1997; Ertl et al., 2020), which frequently appear in bioactive molecules and mediate drug–protein interactions (Mao et al., 2016; He et al., 2010). Functional group-based design is thus a promising direction, explored in recent studies (Zhu et al., 2023; Lin et al., 2023; Nguyen et al., 2024).

However, existing methods generally lack explicit control over functional group numbers, which directly influence binding affinity, selectivity, and in vivo behavior (Li, 2020; Lipinski et al., 1997; Mannhold et al., 2009). Excessive functionalization may enhance molecular activity but often leads to nonspecific binding, whereas fewer functional groups can simplify design and broaden the exploration of chemical space, yet may result in insufficient activity and poor drug-likeness. (Hann et al., 2001; Reynolds et al., 2008; Yang et al., 2010). Moreover, Transformer (Vaswani et al., 2017) also struggles with continuous numerical data such as group counts, as they encode numbers discretely, which limits their ability to capture magnitude, order, and logical relationships—capabilities that

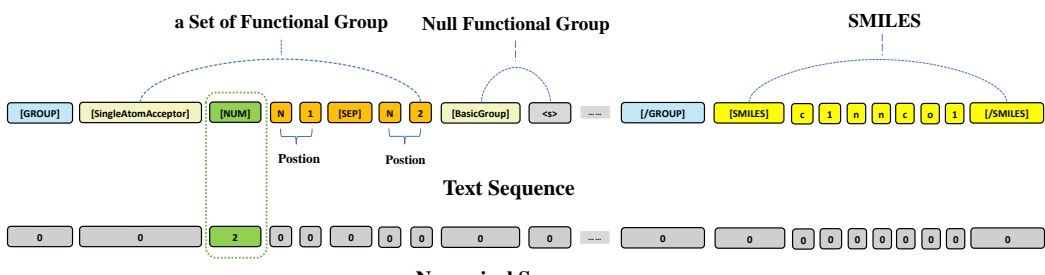

Figure 1: Dual-sequence molecular representation: the top row encodes the text sequence, and the bottom row encodes the functional group quantities. Through this design, we make an explicit distinction between discrete and continuous input signals.

are critical in molecular design (Dziri et al., 2023; Achiam et al., 2023; Thawani et al., 2021; Choi, 2021). This gap highlights the need for models that can seamlessly integrate text and numerical representations of molecular features.

In short, current limitations in drug discovery include: **Lack of Functional Group-Level Modeling and Control.** Most methods do not leverage functional group information or explicitly control their counts, limiting synthesizability and drug-likeness. **Limited Numerical Reasoning in Transformer Models.** Transformers' inductive biases hinder precise control over functional group quantities, reducing their ability to capture variations affecting binding affinity, selectivity, and pharmacological properties.

To address these challenges, we propose NumMolFormer, a numerical reasoning-enhanced transformer for functional-group-based molecular design, with the following contributions:

- **The First Method That Realizes Quantitative Control over Functional Groups.** NumMolFormer introduces a dual-sequence input for functional groups and quantities, an enhanced numerical embedding for magnitude and sign, and a dual-stream differential attention to disentangle textual and numerical signals, enabling precise molecular control beyond standard Transformer limitations.
- **Improved Performance.** NumMolFormer outperforms existing methods and the original Transformer architecture in its numerical understanding of functional groups. Moreover, it can generate molecules with enhanced activity, synthesizability, and drug-likeness based on protein pocket.
- **A Large-Scale Molecular Dataset Accompanied by Functional Group Annotations.** We construct and release a large-scale dataset of 18M molecules, each annotated with counts and positions of 27 functional group types, enabling comprehensive molecular design insights.

## 2 RELATED WORKS

**Transformer in Numerical Task.** xVal (Golkar et al., 2023) introduces a continuous number encoding scheme that represents real numbers using a single token. LUNA (Han et al., 2022) enhances the numerical reasoning and calculation capabilities of transformer-based language models through number augmentation techniques.

**Molecular Design Based on Functional Group.** PGMG (Zhu et al., 2023) encodes seven functional group types and positions, combining graph-based auxiliary information with sequence models to guide molecule generation. DEVELOP (Imrie et al., 2021) integrates graph neural networks and convolutional neural networks to generate molecular structures using three-dimensional pharmacophore information.

**Molecular Design Based on Protein Pocket.** Protein pocket-guided design has become key in structure-based drug discovery. Pocket2Mol (Peng et al., 2022) uses E(3)-equivariant networks to generate molecules compatible with binding pockets, while AutoFragDiff (Ghorbani et al.) employs auto regressive fragment-based diffusion for improved 3D fit.

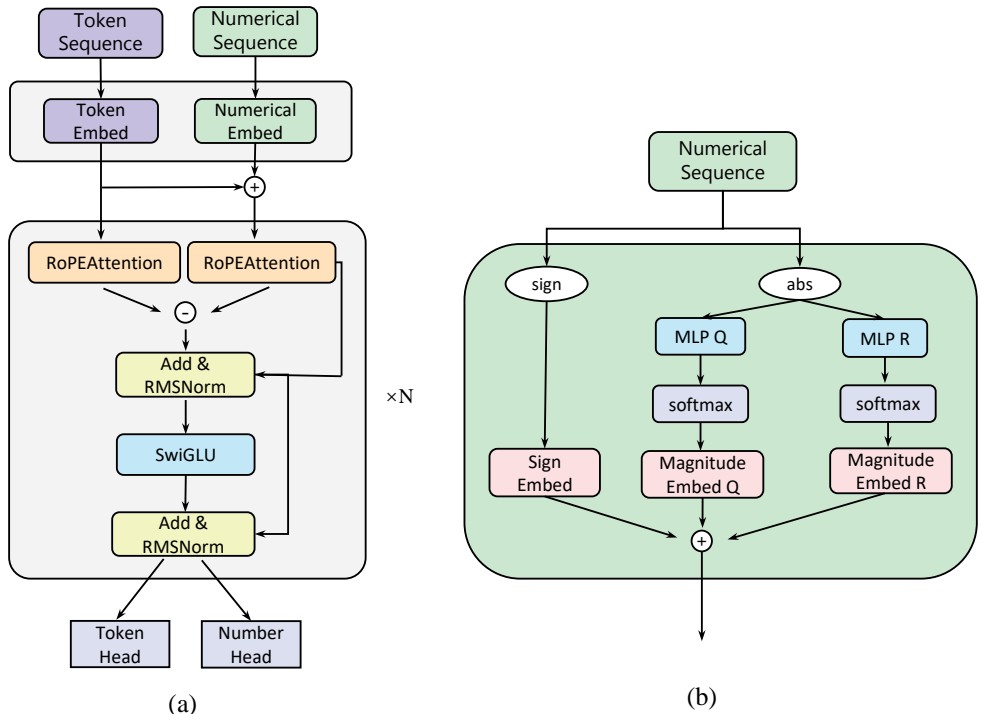

(a)

(b)

Figure 2: Overall architecture of NumMolFormer. (a) NumMolFormer uses a novel representation based on functional group type, count, and position, and incorporates a dual-sequence input, enhanced numerical embedding, and dual-stream differential attention to better model numerical semantics and control functional group counts. (b) The numerical embedding module. It decomposes each value into sign and magnitude components, using separate MLP-based bucketization and sign embedding to capture numerical semantics.

## 3 NUMMOLFORMER

### 3.1 EXTRACTION OF FUNCTIONAL GROUP FEATURES

For all molecules, to ensure consistent and unique representation, we first converte each molecule into a canonical SMILES string using RDKit (Landrum, 2016). To enrich the molecular representation beyond the raw SMILES, we further incorporate domain knowledge in the form of functional group features. Specifically, we leverage RDKit' s predefined feature template file [1] to extract functional groups from the SMILES representation.

In total, we identify 27 types of functional groups, recording both their counts and positions within each molecule (see Appendix E for details on the definition, the complete list of functional groups, and examples).

### 3.2 DUAL-SEQUENCE INPUT STRATEGY

We propose a text-numerical dual-sequence, which is detailed in Figure 1, to disentangle discrete textual semantics from functional group counts, building on extracted functional group information to provide a richer molecular encoding.

Specifically, we generate a textual sequence based on predefined templates, which structurally encodes the functional group type, site indices, and contextual semantics, while simultaneously retaining the original SMILES representation. In parallel, a numerical sequence provides functional group counts strictly aligned with the textual sequence, allowing the model to capture count-based

---

[1] https://github.com/rdkit/rdkit/blob/master/Data/BaseFeatures.fdef

constraints in a continuous and comparable manner. Inspired by xVal (Golkar et al., 2023), we introduce two special tokens, [NUM] and [SEP], to differentiate multiple instances of the same functional group and their distribution across the sequence, establishing an explicit correspondence between the text and numerical signals.

Beyond its fundamental function, this representation provides a richer encoding. It offers complementary information, using text to encode types and context while numbers encode precise counts. The method also helps to mitigate numerical bias, as the continuous numerical input overcomes Transformers' limitations in quantity comparison. Furthermore, this dual-sequence design enhances controllability, helping to enforce specific functional group counts and distributions.

### 3.3 Model Architecture

We present NumMolFormer, a molecular design model that leverages functional group information and improved numerical reasoning. The overall architecture is illustrated in Figure 2.

#### 3.3.1 Numerical Embed

To better capture functional-group counts as structured signals, we design an enhanced numerical embedding module that decomposes each scalar value $n_i$ into three complementary components: (i) a raw numerical injection to preserve the original scale, (ii) a discrete sign embedding to represent directionality, and (iii) a soft magnitude quantization embedding to encode continuous scales. This design allows the Transformer to treat numerical information not as ordinary tokens but as smooth, interpretable, and differentiable features.

**Raw Numerical Injection.** To retain the most direct numerical information, the raw scalar value $n_i$ is broadcast and added to the token embedding:

$$e_i^{\text{raw}} = w_i + n_i \tag{1}$$

where $w_i$ denotes the standard token embedding. This channel ensures that the precise scale of $n_i$ is preserved without discretization.

**Sign Embed.** We further encode the sign of $n_i$ as a discrete categorical feature:

$$s_i = \text{sign}(n_i) \in \{-1, 0, +1\} \quad e_i^{\text{sign}} = \text{Embed}_{\text{sign}}(s_i) \tag{2}$$

where $\text{Embed}_{\text{sign}}$ is a learned embedding layer distinguishing positive, negative, and zero values.

**Magnitude Embed via Dual Soft Magnitude Quantization (SMQ).** The absolute magnitude $|n_i|$ is projected through two parallel lightweight feedforward networks and softly assigned to $K$ learnable bins:

$$p_i^{(q)} = \text{softmax}\left( W_2^{(q)} \, \sigma(W_1^{(q)}|n_i| + b_1^{(q)}) + b_2^{(q)} \right) \tag{3}$$

$$p_i^{(r)} = \text{softmax}\left( W_2^{(r)} \, \sigma(W_1^{(r)}|n_i| + b_1^{(r)}) + b_2^{(r)} \right) \tag{4}$$

where $\sigma$ is the GELU activation. Let $\{v_k^{(q)}\}_{k=1}^K$ and $\{v_k^{(r)}\}_{k=1}^K$ denote two magnitude embedding tables. The final magnitude embedding is given by:

$$e_i^{\text{mag}} = \sum_{k=1}^{K} p_{ik}^{(q)} v_k^{(q)} + \lambda \sum_{k=1}^{K} p_{ik}^{(r)} v_k^{(r)} \tag{5}$$

where $\lambda$ is a learnable scaling parameter.

**Final Numerical Embed.** The overall numerical embedding is the combination of the three channels:

$$e_i^{\text{num}} = e_i^{\text{raw}} + e_i^{\text{sign}} + e_i^{\text{mag}} \tag{6}$$

Compared with naive embedding, our enhanced numerical embedding provides three key benefits in a unified manner: it preserves exact values via raw injection, ensures continuity by smoothly representing scalar variations through SMQ, and enhances interpretability since both the sign and the soft distributions $p_i^{(q)}, p_i^{(r)}$ explicitly reflect numerical structure. This ensures that functional-group counts are modeled as structured and differentiable signals, improving controllability and performance in molecular generation.

### 3.3.2 DUAL-STREAM DIFFERENTIAL ATTENTION

Building on numerical embed, we further introduce a dual-stream differential attention mechanism to disentangle text and numerical contributions. Let $E^{\text{text}}$ denote the text embedding sequence and $E^{\text{fusion}} = E^{\text{text}} + E^{\text{num}}$ the fused embedding sequence. Self-attention is computed in parallel for the two streams:

$$H^{\text{text}} = \text{Attention}(E^{\text{text}}, E^{\text{text}}, E^{\text{text}}) \tag{7}$$

$$H^{\text{fusion}} = \text{Attention}(E^{\text{fusion}}, E^{\text{fusion}}, E^{\text{fusion}}) \tag{8}$$

The incremental effect of numerical features is then isolated via subtraction:

$$H^{\text{diff}} = H^{\text{fusion}} - H^{\text{text}} \tag{9}$$

This differential signal is integrated with the fused embeddings through residual connection and normalization,

$$E' = \text{Norm}\big(E^{\text{fusion}} + H^{\text{diff}}\big) \tag{10}$$

followed by a SwiGLU feedforward transformation and a second normalization layer.

### 3.4 UNCONDITIONAL MOLECULAR PRETRAINING

Currently, High-quality experimental data for protein-ligand complexes remain extremely scarce. The largest dataset, PDBbind (Liu et al., 2017), contains fewer than 20,000 complexes. Even when considering augmented datasets generated using docking software, such as CrossDocked2020 (Francoeur et al., 2020), the total number of complexes only reaches approximately 100,000. Such limited data are insufficient to train a robust machine learning model.

To address this, we adopt a pretraining-finetuning strategy: the model is first trained on a large-scale molecular dataset, followed by finetuning on protein-ligand complex data, enabling conditional molecular generation capabilities. Specifically, we obtain approximately 18 million raw SMILES molecules from Uni-Mol (Lu et al., 2024; Zhou et al., 2023). The functional groups of these molecules are annotated following the extraction procedure described in Section 3.1.

To enable our dual-channel autoregressive model to capture both functional group and molecular sequence information, as well as the corresponding functional group counts, we employ a composite loss function. The loss jointly predicts the next textual token in the sequence and the associated numerical value representing functional group counts. Formally, let $\hat{s}_{\text{tok}}$ denote the predicted token sequence, $s_{\text{tok}}$ the ground-truth token sequence, $\hat{s}_{\text{num}}$ the predicted numerical sequence, and $s_{\text{num}}$ the ground-truth numerical sequence. The loss is defined as:

$$\mathcal{L}(\hat{s}, s) = \text{CE}(\hat{s}_{\text{tok}}, s_{\text{tok}}) + \text{MSE}(\hat{s}_{\text{num}}, s_{\text{num}}) \tag{11}$$

where $\text{CE}(\cdot)$ denotes the cross-entropy loss and $\text{MSE}(\cdot)$ denotes the mean squared error loss.

### 3.5 FINE-TUNING ON PROTEIN POCKET CONDITIONS

#### 3.5.1 SELF-SUPERVISED FINE-TUNING

At this stage, the model input is augmented with amino acid sequence information from the protein binding pocket. We freeze ESM2 (650M) (Rives et al., 2019; Lin et al., 2022) to extract informative representations from protein sequences, which are then encoded as feature embeddings. These embeddings are integrated into the model's embedding layer via a cross-attention mechanism, allowing the network to capture contextual interactions between the protein environment and molecular representations. The resulting combined embeddings are trained using the auto regressive objective defined in Equation 11.

### 3.5.2 REINFORCEMENT LEARNING FINE-TUNING

Inspired by reinforcement learning (RL)-based molecular generation methods (Olivecrona et al., 2017; Hu et al., 2025), an RL agent with the NumMolFormer architecture is initialized with self-supervised weights. For each protein pocket, a molecular property scoring function is used as the RL reward, and the agent is iteratively optimized to maximize the expected reward. At each RL step, the agent samples a batch of ligands, and the regularized maximum likelihood estimation (MLE) loss (Gummesson Svensson et al., 2024) of each ligand is computed to update the agent:

$$L_{\text{design}}\left(\hat{s}^{\text{lig}}\right) = \left( \log \pi_{\text{pre-trained}}\left(\hat{s}_{\text{tok}}^{\text{lig\_smiles}}\right) + \sigma \cdot R(m) - \log \pi_{\text{agent}}\left(\hat{s}_{\text{tok}}^{\text{lig\_smiles}}\right) \right)^2 \quad (12)$$

where $\hat{s}^{\text{lig}}$ is a generated ligand, $m$ its corresponding molecule, $R(m)$ the reward evaluating molecular properties, $\pi_{\text{pre-trained}}$ and $\pi_{\text{agent}}$ the likelihoods of the pre-trained model and agent, and $\sigma$ a scaling hyperparameter.

The reward function combines docking, drug-likeness, and synthetic accessibility scores:

$$R(m) = 0.4 \cdot f_{\text{dock}}(S_{\text{dock}}(m)) + 0.3 \cdot f_{\text{qed}}(\text{QED}(m)) + 0.3 \cdot f_{\text{sa}}(\text{SA}(m)) \quad (13)$$

with normalized scoring functions:

$$f_{\text{dock}}(x) = \frac{1}{1 + e^{-k(x+3.0)}}, \quad f_{\text{qed}}(x) = \frac{1}{1 + e^{-k(x-0.2)}}, \quad f_{\text{sa}}(x) = \frac{1}{1 + e^{-k(x-0.5)}} \quad (14)$$

where $S_{\text{dock}}(m)$, $\text{QED}(m)$, and $\text{SA}(m)$ denote the docking, drug-likeness, and synthetic accessibility scores of $m$, with $k = 10$.

## 4 EXPERIMENT

This section presents the results of unconditional and protein-conditioned molecular generation. Detailed training configurations and hyperparameters are provided in Appendix A, B. Moreover, ablation study can be found in Appendix C.2.

### 4.1 UNCONDITIONAL DRUG DESIGN

**Data.** We randomly partition the 18 million pre-training dataset into training, validation, and test sets with ratios of 0.9998/0.0001/0.0001, respectively, and designate the first 2,048 molecules (Devlin et al., 2019) from the test set as the actual evaluation set.

**Baselines.** To the best of our knowledge, no existing molecular generation method is capable of achieving precise quantitative control over functional groups. To assess this capability, we evaluate several general-purpose large models (GPT-4.1 (Fachada et al., 2025), DeepSeek-v3.1 (Liu et al., 2024), Grok-4 (xAI, 2025)), and additionally retrain a LLaMA model (Touvron et al., 2023) of comparable scale on our dataset to serve as a baseline (for more retrain details, refer to Appendix C.1).

**Molecule Quality Evaluation.** We adopt widely used metrics to evaluate the quality of generated molecules. Specifically, **Validity** measures the proportion of generated SMILES strings corresponding to chemically valid molecules. **Uniqueness** quantifies the fraction of distinct molecules, i.e., not duplicated, while **Novelty** indicates the proportion of molecules not present in the training set. **Lipinski** (Lipinski et al., 1997) reports the fraction of molecules satisfying all five of Lipinski's rules for drug-likeness. Additionally, **QED** (Quantitative Estimate of Drug-likeness) (Bickerton et al., 2012) evaluates overall drug-likeness on a continuous scale, and **SA** (Synthetic Accessibility) (Ertl & Schuffenhauer, 2009) measures the ease of chemical synthesis.

As illustrated in Table 1, the molecules we generated exhibit superior drug-likeness and synthetic accessibility, while significantly outperforming existing large models and traditional Transformer-based LLaMA architectures in terms of functional group constraints.

**Functional Group Constraint Evaluation.** To evaluate the model's ability to satisfy functional group count constraints, we first compute the **Mean Squared Error (MSE)** by extracting counts of 27 functional groups from each generated molecule and comparing them to the target distributions. We also measure the **Exact Match Rate (EMR)**, which indicates the fraction of molecules whose functional group counts exactly satisfy the specified constraints. Finally, the **Fuzzy Match Rate (FMR)** quantifies the proportion of molecules whose counts deviate from the target by at most $\pm 3$. As illustrated in Table 1, Our model achieves better functional group count constraints, successfully generating a high percentage of molecules that adhere to both strong and weak constraints. This performance substantially surpasses baseline methods.

Table 1: The table compares existing large models (GPT-4.1, Grok-4, DeepSeek-v3.1), a LLaMA model retrained on our dataset, and our proposed NumMolFormer. Metrics with an upward arrow ($\uparrow$) indicate higher values are better, while metrics with a downward arrow ($\downarrow$) indicate lower values are better. For each metric, the best result is highlighted in **bold**.

| Metrics | Existing Large Models | | | Retrained | Our |
|---|---|---|---|---|---|
| | GPT-4.1 | Grok-4 | DeepSeek-v3.1 | LLaMA | |
| **General Properties** | | | | | |
| Validity ($\uparrow$) | 0.612 | 0.080 | **0.954** | 0.347 | 0.738 |
| Uniqueness ($\uparrow$) | 0.981 | 0.998 | 0.884 | 0.994 | **1.000** |
| Novelty ($\uparrow$) | 0.980 | 0.998 | 0.985 | 0.998 | **1.000** |
| Lipinski ($\uparrow$) | 0.514 | 0.750 | 0.812 | 0.923 | **0.938** |
| QED ($\uparrow$) | 0.590 | 0.539 | 0.628 | 0.631 | **0.644** |
| SA ($\uparrow$) | 0.794 | 0.747 | **0.882** | 0.763 | 0.800 |
| **Functional Group Match** | | | | | |
| Mean Squared Error ($\downarrow$) | 0.553 | 0.412 | 0.519 | 0.290 | **0.121** |
| Exact Match Rate ($\uparrow$) | 0.006 | 0.000 | 0.005 | 0.106 | **0.362** |
| Fuzzy Match Rate ($\uparrow$) | 0.217 | 0.187 | 0.208 | 0.361 | **0.814** |

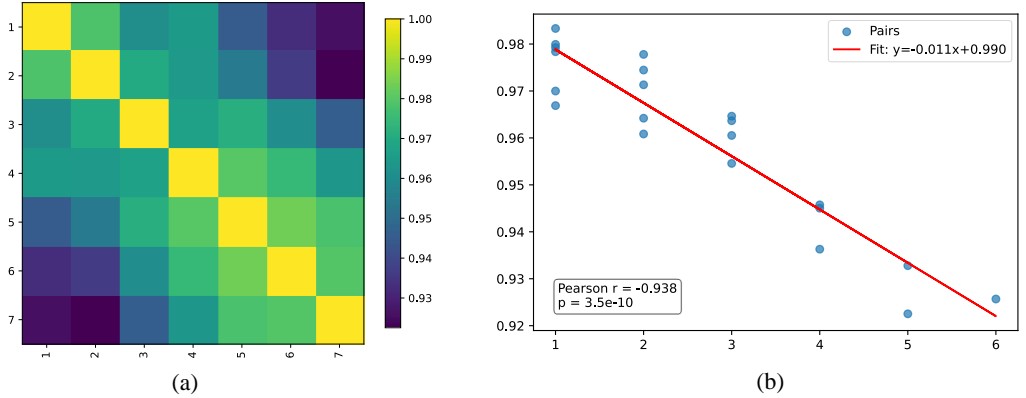

(a)  (b)

Figure 3: Analysis on the Numerical Sensitivity of NumMolFormer. (a) Cosine similarity between model output embeddings for inputs with different numbers of functional groups. (b) Relationship between the difference in functional group count and the cosine similarity of their embeddings.

**Numerical Sensitivity Evaluation.** To quantitatively evaluate the numerical sensitivity of the model, we select seven representative sequences with the largest gradients from the test set. In each sequence, all functional group quantities are held constant except for *SingleAtomAcceptor*, whose count is varied from 1 to 7. For every sequence, we extract the final molecular embedding produced by our model. As shown in Figure 3, (a) embeddings gradually become less similar as the number of input functional groups increases, and (b) functional group count differences are strongly negatively correlated with embedding similarity, with a Pearson correlation coefficient of $-\mathbf{0.93}$.

These results indicate that greater differences in functional group counts lead to lower molecular embedding similarity, demonstrating the model's sensitivity to functional group variations.

## 4.2 POCKET-AWARE DRUG DESIGN

Table 2: Experimental results of NumMolFormer and other baselines on pocket-aware drug design, following the results reported in DecompDiff (Guan et al., 2024) and D3FG (Lin et al., 2023). (↑) / (↓) denotes that a higher / lower value is better. The best result in each column is **bolded**.

| Methods | Vina Score (↓) | Vina Dock (↓) | QED (↑) | SA (↑) | Success Rate (↑) |
|---|---|---|---|---|---|
| Reference | -6.36 | -7.45 | 0.48 | 0.73 | 25.0% |
| AR | -5.75 | -6.75 | 0.51 | 0.63 | 7.1% |
| liGAN | - | -6.33 | 0.39 | 0.59 | 3.9% |
| GraphBP | - | -4.80 | 0.43 | 0.49 | 0.1% |
| Pocket2Mol | -5.14 | -7.15 | 0.56 | 0.74 | 24.4% |
| TargetDiff | -5.47 | -7.80 | 0.48 | 0.58 | 10.5% |
| DecompDiff | -5.67 | **-8.39** | 0.45 | 0.61 | 24.5% |
| D3FG | - | -7.19 | 0.482 | 0.731 | - |
| NumMolFormer | **-6.08** | -7.84 | **0.63** | **0.75** | **42.24%** |

**Data.** Following previous works (Peng et al., 2022; Guan et al., 2023b), we select 100 protein pockets from the CrossDocked2020 (Francoeur et al., 2020) dataset that exhibit low similarity ($< 30\%$) to the protein sequences of pocket-ligand complexes used in pre-training, leading to about 100,000 pairs of pocket-ligand complexes, with 100 novel complexes as references for evaluation.

**Baselines.** We compare NumMolFormer against various baselines for pocket-aware molecular generation, including AR (Luo et al., 2021), liGAN (Ragoza et al., 2022), GraphBP (Liu et al., 2022a), Pocket2Mol (Peng et al., 2022), TargetDiff (Guan et al., 2023a), DecompDiff (Guan et al., 2024), D3FG (Lin et al., 2023). For more details, please refer to Appendix D.1.

**Experimental Setup.** In alignment with previous works (Peng et al., 2022; Guan et al., 2024), we evaluate 100 molecules generated for each protein pocket, resulting in a total of 10,000 complex pairs for analysis.

Table 3: Ratio of the top six functional groups with the highest frequency in Crossdocked2020. Ref is calculated in the training set. MSE is obtained between rows of Ref and different methods' Ratio. (↓) denotes that a lower value is better. The best result in each column is **bolded**.

| Functional Group | Ref. | Pocket2Mol | TargetDiff | DiffSbdd | D3FG | Ours |
|---|---|---|---|---|---|---|
| c1ccccc1 | 0.712 | 0.583 | 0.293 | 0.131 | 0.608 | **0.643** |
| NC=O | 0.266 | 0.089 | 0.149 | 0.010 | 0.159 | **0.328** |
| c1ccncc1 | 0.082 | 0.086 | 0.052 | 0.001 | **0.078** | 0.136 |
| c1ncc2nc[nH]c2n1 | 0.061 | 0.001 | 0.000 | 0.000 | **0.030** | 0.000 |
| OCO | 0.034 | **0.024** | 0.097 | 0.001 | 0.075 | 0.090 |
| c1cncnc1 | 0.032 | 0.010 | **0.015** | 0.000 | 0.013 | 0.011 |
| MSE ( ↓ ) | - | 0.0087 | 0.0330 | 0.0692 | 0.0042 | **0.0031** |

**Generated Molecule Evaluation.** As shown in Table 2, molecules generated by NumMolFormer consistently outperform all baselines across five key metrics: **Vina Score** (binding affinity), **Vina Dock** (binding affinity), **QED** (drug-likeness), **SA** (Synthetic Accessibility), and **Success Rate** (Long et al., 2022; Guan et al., 2024) (proportion of molecules satisfying Vina Dock $< -8.18$, QED $> 0.25$, and SA $> 0.59$). Remarkably, NumMolFormer even exceeds the dataset reference values on all metrics, highlighting its effectiveness in multi-objective molecular optimization.

**Functional Group Evaluation.** We assess the functional group distributions of generated molecules against the training set reference (Ref) and baselines (see Appendix D.2 for more details). This provides insight into the model's ability to capture underlying chemical patterns. As shown in Table 3, our method accurately reproduces high-frequency functional groups, outperforming previous approaches, while also reasonably reflecting low-frequency groups, demonstrating both effective and chemically consistent molecular generation.

**Docking Case.** Table 4 compares the reference molecule with molecules generated by our model for the pocket `ACE_HUMAN_650_1230_0` and `NEP_HUMAN_54_750_0`, reporting Vina Dock, QED, and SA. The generated molecules show improved docking scores and comparable drug-likeness and synthetic accessibility. Figure 4 shows their docking poses in the binding pocket.

Table 4: Comparison of the reference molecule and molecules generated by our model for the pockets `ACE_HUMAN_650_1230_0` and `NEP_HUMAN_54_750_0`, including Vina Dock, QED, and SA.

| Molecule | SMILES | Vina Dock ($\downarrow$) | QED ($\uparrow$) | SA ($\uparrow$) |
|---|---|---|---|---|
| | `ACE_HUMAN_650_1230_0` | | | |
| Reference | C[C@H](NC(=O)O)C(=O)O | -3.6 | 0.483 | 0.820 |
| Generated 1 | Cc1coc(NCNc2cc(C)ccc2)=N1 | **-7.2** | 0.773 | 0.813 |
| Generated 2 | Cc1ncc(CCNc4cc(C)ncc4)o1 | -6.1 | **0.854** | **0.839** |
| Generated 3 | Nc1coc(CCNc2cc(C)ncc2)=N1 | -6.0 | 0.815 | 0.793 |
| | `NEP_HUMAN_54_750_0` | | | |
| Reference | C[C@H](NC(=O)[C@H](Cc1ccc(-c2ccccc2)cc1)C[P@@](=O)(O)[C@H](C)N)C(=O)O | -5.7 | 0.463 | 0.731 |
| Generated 1 | CC(NC=C(Cc1ccc(-c2ccccc1)c(O)c2)C(=O)O)=O | **-8.9** | 0.746 | 0.758 |
| Generated 2 | CC(NC(=O)C(Cc1ccc(-c2ccccc1)cc2)C(=O)O)=O | -6.9 | 0.848 | 0.745 |
| Generated 3 | CC(NC=C(Cc1ccc(-c2ccccc1)cc2)C(=O)O)=O | -6.4 | **0.852** | **0.767** |

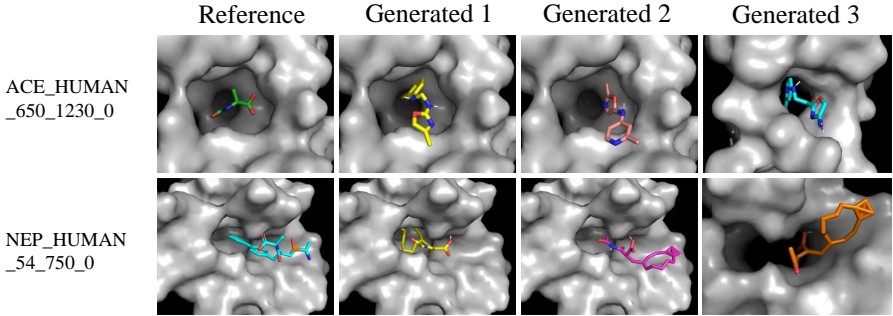

Figure 4: Docking results of the reference and generated molecules in `ACE_HUMAN_650_1230_0` and `NEP_HUMAN_54_750_0` pocket. Generated molecules outperform the reference compounds because they preserve the key functional groups and aromatic/ hydrophobic fragments necessary for protein binding, while removing unnecessary stereocenters, highly polar groups. The structural simplification enhances binding affinity, improves drug-likeness and synthetic accessibility.

## 5 CONCLUSION AND DISCUSSION

As the first molecular generation model to achieve quantitative control over functional groups, Num-MolFormer introduces a new method for molecular design. By effectively encoding functional group information, the model generates high-quality molecules that satisfy numerical constraints, significantly enhancing their drug-likeness, synthesizability, and biological activity. While the model faces limitations, like its reliance on a fixed set of functional group types and unvalidated performance on a larger scale, it lays a crucial foundation for achieving precise and controllable molecular generation.

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

## A  TRAINING DETAILS

**Pretrain.**  We conducted training on the full dataset for one epoch using four Nvidia L20 GPUs, with a total training time of less than 24 hours. The AdamW optimizer (Loshchilov & Hutter, 2017) was employed with a learning rate of $3 \times 10^{-4}$ and a weight decay of 0.01. The batch size was set to 36, with gradient accumulation performed over 5 steps, resulting in an effective batch size of 720. A warmup phase of 0.25 was applied, followed by cosine learning rate decay. Notably, mixed-precision training with 16-bit floating point was not adopted, as it led to instability during training. We hypothesize that this instability arises from the high numerical sensitivity of our model architecture.

**Self-supervised strategy training.**  These settings are largely consistent with those used during the pretraining phase. However, due to the reduced size of the training dataset, we adjust the warmup ratio to 0.2, set the learning rate to $2 \times 10^{-4}$, and remove gradient accumulation.

**RL Finetuning**  We perform 200 reinforcement learning (RL) steps for each protein pocket, with a batch size of 32 and a fixed learning rate of $1 \times 10^{-4}$. The parameter $\sigma$ in Eq. 12 is set to 100. Each RL process takes less than 2 hours on a single Nvidia L20 GPU with 128 CPU cores, while the computation of the Vina Dock reward is carried out in parallel on the CPU cores.

## B  MODEL DETAILS

To investigate the impact of different architectural choices on model performance, we conducted a hyperparameter exploration varying the number of layers, attention heads, embedding dimensions, and magnitude parameters. Table 5 summarizes the results, reporting the corresponding training loss for each configuration. The best-performing configuration, highlighted in **bold**, consists of 10 layers, 12 attention heads, an embedding size of 768, and a magnitude of 200, achieving the lowest training loss of 0.171. These results guided our selection of the final model architecture for subsequent experiments.

Table 5: Exploration of Model Hyperparameters

| Layer | Head | Embedding length | K | Loss |
|-------|------|------------------|-----|-------|
| 12 | 12 | 768 | 200 | 0.176 |
| **10** | **12** | **768** | **200** | **0.171** |
| 10 | 12 | 768 | 100 | 0.192 |
| 8 | 12 | 768 | 200 | 0.198 |
| 8 | 8 | 768 | 100 | 0.216 |
| 8 | 8 | 256 | 100 | 0.243 |

## C  UNCONDITIONAL DRUG DESIGN SUPPLEMENT

### C.1  RETRAIN LLAMA

**LLaMA Squence.**  Since the LLaMA architecture cannot directly handle parallel sequences, we reconstructed the input sequences to preserve functional group information while making them compatible with the model, as illustrated in Figure 5.

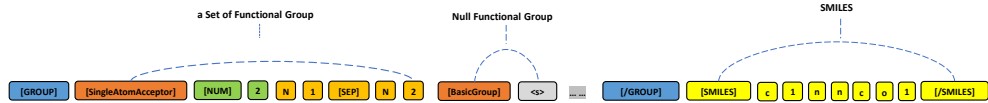

Figure 5: the sequences used for LLaMA pre-training

**Numerical Injection in LLaMA.** As shown in the Figure 5, we inject numerical values in textual form by reconstructing the input sequence, which is consistent with the conventional approach. For a fairer comparison, it should be noted that the LLaMA model also explicitly incorporates numerical information via Equation 1.

**Training Details.** This part is fully consistent with the training details of NumMolFormer.

## C.2 ABLATION STUDY.

We compared mainstream decoder-only architectures LLaMA with the NumMolFormer architecture and conducted ablation studies to analyze the impact of its individual components on performance. Specifically, **w/o embed** refers to retaining only Equation 1 while removing all other components in Numerical Embed. **w/o attention** denotes the attention module without Equation 7 and Equation 9, and Equation 10 uses only $E^{\text{fusion}}$. **w/o all** applies both of the aforementioned modifications simultaneously.

As shown in Table 6, the embedding module mainly improves molecular validity and reduces MSE, indicating its key role in encoding numerical features accurately. The attention module also contributes to validity and MSE, reflecting its importance in capturing interactions between molecular representations. Removing both modules leads to the largest degradation, confirming that embedding and attention provide complementary benefits: embedding ensures correct feature representation, while attention models inter-feature dependencies. This ablation highlights that both components are crucial for generating chemically valid and accurate molecules.

Table 6: Ablation studies comparing NumMolFormer with widely adopted open-source model architectures LLaMA. ($\uparrow$) / ($\downarrow$) denotes that a higher / lower value is better. The best result in each column is **bolded**.

| Methods | Validity ($\uparrow$) | Uniqueness ($\uparrow$) | Novelty ($\uparrow$) | Lipinski ($\uparrow$) | MSE ($\downarrow$) |
|---|---|---|---|---|---|
| LLaMA | 0.426 | 1.0 | 1.0 | **0.974** | 0.283 |
| Ours | **0.738** | **1.0** | **1.0** | 0.927 | **0.121** |
| Ours w/o Emb. | 0.635 | 1.0 | 1.0 | 0.933 | 0.260 |
| Ours w/o Attn. | 0.628 | 1.0 | 1.0 | 0.927 | 0.240 |
| Ours w/o All | 0.583 | 1.0 | 1.0 | 0.943 | 0.305 |

## C.3 DISTRIBUTION ANALYSIS

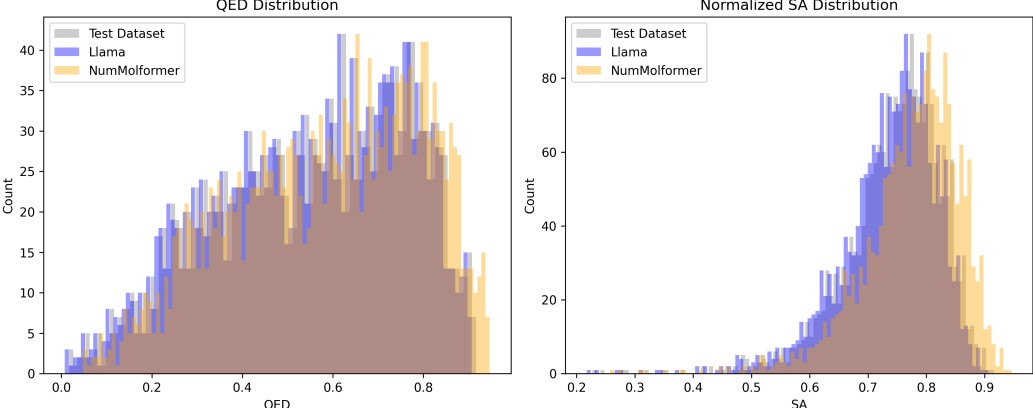

Figure 6: This figure illustrates the distributions of QED and SA for molecules in the test set generated by LLaMA and NumMolFormer. It can be observed that NumMolFormer exhibits superior distributions for both metrics, indicating a better capability in molecular representation.

## D  POCKET-AWARE DRUG DESIGN SUPPLEMENT

### D.1  MODEL COMPARISON

**Pocket2Mol and GraphBP** are auto-regressive models that generate molecules at the level of individual atoms. These methods follow the classical atom-bond paradigm, sequentially constructing molecules by adding one atom at a time and forming bonds with previously placed atoms. **Target-Diff and DecompDiff** are diffusion-based methods. Although they can capture global information, both model molecules solely at the level of atom types and bond types. **AR** introduced a 3D generative model that directly generates molecular conformations conditioned on protein structures, enabling end-to-end design of molecules with favorable binding properties. **liGAN** proposes a deep generative model that generates 3D molecules conditioned on receptor binding sites, capturing both geometric and chemical constraints to produce realistic ligand structures. **D3FG** is a functional-group-based diffusion model that decomposes molecules into rigid functional groups and linkers, and generates new molecules through a diffusion and denoising process.

### D.2  TABLE 3 SUPPLEMENT

The comparative data were obtained from D3FG (Lin et al., 2023). It is noteworthy that D3FG defines functional groups based on fragment occurrence frequencies, which is inconsistent with our functional group definitions derived from biochemical priors. Therefore, we removed four fragments from D3FG—`c1cn[nH]c1`, `O=P(O)(O)O`, `NS(=O)=O`, and `O=CO`—as these fragments are not included in our functional group definitions.

## E  FUNCTIONAL GROUP FEATURES

To ensure feature consistency, when a functional group was absent, we set its count to zero and used a placeholder token `[SEP]` for the positional information. This padding strategy guaranteed strict alignment across all samples and enabled efficient large-scale batch training.

Formally, for a given molecule $M$, the functional-group set is represented as

$$F = \{(c_i, n_i, p_i)\}_{i=1}^T \tag{15}$$

where $c_i$ denotes the $i$-th functional group type, $n_i$ its corresponding count, and $p_i$ a structured representation of its positions within the molecule.

$$p_i = \{P_i^{(1)}, P_i^{(2)}, \ldots, P_i^{(k_i)}\}, \quad P_i^{(j)} = \{a_1^{(j)}, a_2^{(j)}, \ldots, a_{m_j}^{(j)}\} \tag{16}$$

where $k_i$ is the number of distinct substructures in which the functional group appears, $m_j$ is the number of atoms in the $j$-th substructure, and $a_l^{(j)}$ denotes the atom type and index within the molecule.

Table 7 lists the 27 functional groups extracted from RDKit.

### E.1  FUNCTIONAL GROUP STATISTICS

Figure 7 illustrates the proportion of molecules containing each functional group in the training set. The result reveals a significant variation in the occurrence rates of these groups. **SingleAtomAcceptor** and **Arom6** are the most frequently observed, with occurrence ratios of 96.33% and 82.55%, respectively, indicating their common presence in the studied molecules. Other high-frequency groups, with proportions over 5%, include **SingleAtomDonor** (81.18%), **BasicGroup** (15.25%), **Arom8** (6.06%), **ChainTwoWayAttach** (42.66%), **tButyl** (7.23%), and **iPropyl** (7.20%). Conversely, many functional groups appear with very low frequencies. For instance, **AcidicGroup** has a frequency of 4.84%, while several others like **PosN** and **RH6_6** are found in less than 1% of the molecules. Notably, a significant number of groups, including all **ZnBinder3-6** and **RH6_5**, **RH4_4**, and **RH3_3**, have an occurrence frequency of 0.00%, suggesting they are absent from this specific set of molecules. This data provides valuable insight into the distribution and importance of different functional groups within the analyzed molecular set.

Table 7: List of functional groups used in our study.

| Category | Functional groups |
|---|---|
| Single-atom donors/acceptors | Single-atom hydrogen bond donors
Single-atom hydrogen bond acceptors |
| Acid-base groups | Acidic groups
Basic groups |
| Aromatic groups | Arom4
Arom5
Arom6
Arom7
Arom8 |
| Zinc-binding groups | ZnBinder1
ZnBinder2
ZnBinder3
ZnBinder4
ZnBinder5
ZnBinder6 |
| Branching groups | ThreeWayAttach
ChainTwoWayAttach |
| Common substituents | Nitro2
tButyl
iPropyl |

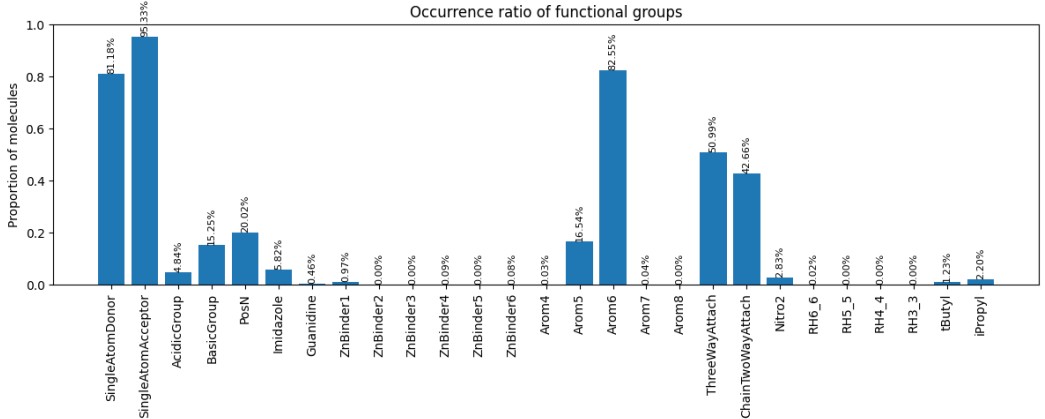

Figure 7: Occurrence Frequencies of 27 Functional Groups.

E.2 VISUALIZATIONS OF 27 FUNCTIONAL GROUPS IN MOLECULES.

Figures 8 presents visualizations of functional group structures, with each group highlighted by circles of different colors. The dataset does not contain the following four functional groups: Zn-Binder2, ZnBinder3, ZnBinder5, and Arom8; therefore, they are not shown.

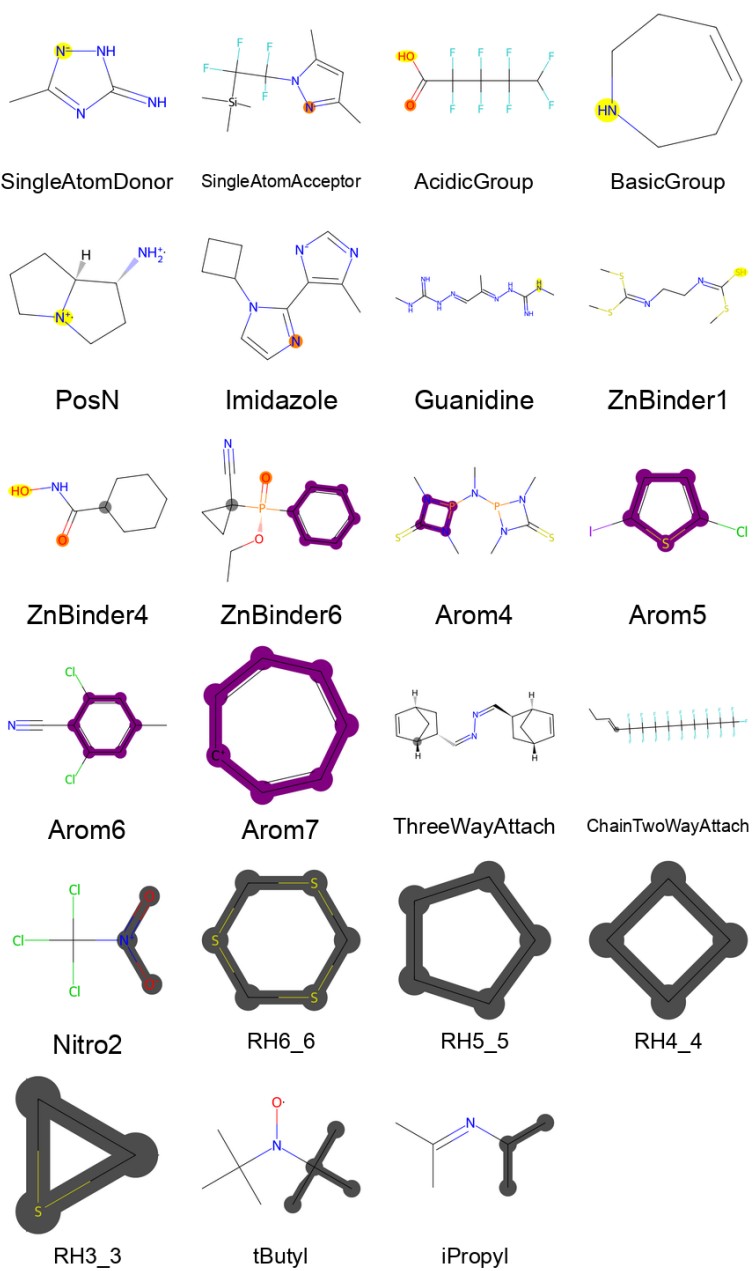

Figure 8: Visualization of functional groups in molecules.

