# OpenReview forum: "NumMolFormer: Enhancing Transformer Numerical Reasoning for Functional-Group-Based Molecule Generation"
_ICLR.cc/2026/Conference — ICLR 2026 Conference Withdrawn Submission_

### Official Review · Reviewer_nQcB · 2025-10-27

**Soundness:** 2
**Presentation:** 2
**Contribution:** 2
**Rating:** 4
**Confidence:** 3

**Summary:**

This paper proposes NumMolFormer,  a novel molecular generation method that integrates functional group knowledge with numerical modeling. NumMolFormer employs a dual-sequence representation that jointly encodes text sequence tokens of functional groups with their quantitative information. To overcome data scarcity, they also build an 18 million molecule dataset with functional group annotations for pretraining. Experimental results demonstrate that NumMolFormer can effectively control functional groups in molecular generation.

**Strengths:**

- This work releases a large-scale dataset of 18M molecules, each annotated with counts and positions of 27 functional group types. This dataset could be useful for future research in this area.
- This paper has good contribution in quantitative control over functional groups, and outperforms existing methods in its numerical understanding of functional groups.

**Weaknesses:**

- It’s better to also report diversity and Lipinski metric in Table 2. Also, generation efficiency is also an important factor when evaluating the practically usefulness of the model.
- Some functional group based generation methods are not discussed in this paper, such as [1][2]
- This method relies on a fixed set of functional group types, which may limits generalizability and may bias the model toward common groups only.

[1] Zhang, Zaixi, et al. "Molecule generation for target protein binding with structural motifs." ICLR \
[2] Fu, Cong, et al. "Fragment and geometry aware tokenization of molecules for structure-based drug design using language models." ICLR

**Questions:**

Please refer to the weakness part.

---

### Official Review · Reviewer_jcnF · 2025-11-01

**Soundness:** 2
**Presentation:** 2
**Contribution:** 1
**Rating:** 2
**Confidence:** 4

**Summary:**

The authors represent a molecule as a pair of text and numerical sequences, augmenting SMILES with numerical annotations of how many of each functional group appears in the molecule. To encode these numerical values in a transformer, NumMolFormer introduces numerical embeddings of the raw number, its sign, and its magnitude, as well as separate attention layers for discrete and numerical sequences. They pretrain on a large dataset of molecules, fine-tune on protein-ligand complexes, and additionally perform reinforcement learning to optimize docking, drug-likeness, and synthetic accessibility. Results show that NumMolFormer achieves better functional group control and designs better drugs than LLMs without numerical encodings.

**Strengths:**

To the best of my knowledge, the authors propose novel numerical encodings of the raw value, sign, and soft-quantized magnitude of numbers.

The authors perform a detailed analysis of how NumMolFormer's numerical embeddings achieve better control of functional groups than discrete-only LLMs.

**Weaknesses:**

The proposal of numerical embeddings *specifically* for controlling the amount of each functional group is poorly motivated. The authors emphasize that embedding the number of functional groups requires a continuous, differentiable encoding, yet the number of functional groups is a discrete but ordinal value - Figure 3b shows that this number only takes on discrete values of 1-7. The number of functional groups also cannot be negative, which leaves me confused about why the sign of the number needs to be encoded. I would understand if numerical encodings are used to embed property values or 3D coordinates, but this work does not mention these use-cases.

The work is not placed in the context of literature, and is of limited novelty. This work is very similar to 3DMolFormer [1] in its ideas, execution, and presentation but 3DMolFormer is not cited in the related works section.

- Ideas: both works propose a dual-sequence transformer with cross-entropy + MSE loss
- Execution: same experiments on pocket-conditional design
- Presentation: similar language
  - 3DMolFormer: "Inspired by 1D RL-based molecular generation methods (Olivecrona et al., 2017), an RL agent with
the 3DMolFormer architecture is initialized with the pre-trained weights..."
  - NumMolFormer: "Inspired by reinforcement learning (RL)-based molecular generation methods (Olivecrona et al.,
2017; Hu et al., 2025), an RL agent with the NumMolFormer architecture is initialized with self-
supervised weights."

The related works section also does not cite any papers from 2024 or 2025, missing several recent works on numerical embeddings [2], fragment-based generative design [3,4], and pocket-based design [5].

The need for high functional group match rates is not well motivated. Table 1 shows that baselines without numerical encodings, and even non-finetuned LLMs, can achieve appreciable functional group match rates. Checking for functional groups has only nominal computational cost, so if the goal is to generate molecules with a specific functional group, it seems much simpler to just ask GPT-4.1 to generate more molecules and then filter, potentially with evolutionary strategies.

The experimental setup for pocket-conditional design is not explained in the paper. Does the model take as input the atomic coordinates of the pocket, or just the full protein sequence's ESM2 embeddings?

Experiments on pocket-conditional design also do not compare to strong baselines such as 3DMolFormer, which would outperform NumMolFormer.

The GitHub link breaks anonymity of authors.

[1] Hu, X., Liu, G., Chen, C., Zhao, Y., Zhang, H., & Liu, X. (2025). 3DMolFormer: A Dual-channel Framework for Structure-based Drug Discovery. arXiv preprint arXiv:2502.05107.

[2] McLeish, S., Bansal, A., Stein, A., Jain, N., Kirchenbauer, J., Bartoldson, B., ... & Goldstein, T. (2024). Transformers can do arithmetic with the right embeddings. Advances in Neural Information Processing Systems, 37, 108012-108041.

[3] Xie, W., Zhang, J., Xie, Q., Gong, C., Ren, Y., Xie, J., ... & Pei, J. (2025). Accelerating discovery of bioactive ligands with pharmacophore-informed generative models. Nature communications, 16(1), 2391.

[4] Lee, S., Kreis, K., Veccham, S. P., Liu, M., Reidenbach, D., Peng, Y., ... & Vahdat, A. (2025). Genmol: A drug discovery generalist with discrete diffusion. arXiv preprint arXiv:2501.06158.

[5] Wang, J., Luo, H., Qin, R., Wang, M., Wan, X., Fang, M., ... & Kang, Y. (2025). 3DSMILES-GPT: 3D molecular pocket-based generation with token-only large language model. Chemical Science, 16(2), 637-648.

**Questions:**

1. Why are continuous numerical embeddings required for discrete but ordinal values of the number of functional groups?
2. The paper should provide more context in related work and justify its contributions in comparison to 3DMolFormer. The paper should also include the results of 3DMolFormer in Table 2.
3. It would be interesting to encode 3D coordinates and properties with NumMolFormer.
4. For pocket-conditional design, does the model take as input the atomic coordinates of the pocket, or just the full protein sequence's ESM2 embeddings?
5. The paper claims that differentiable numerical encodings are important, yet there are no experiments that differentiate through these numerical encodings. An interesting experiment could be to control functional group counts by differentiating through these encodings.
6. The paper should provide more motivation on why it is necessary to control functional group counts with a tailored architecture, when discrete-only LLMs are able to perform this task and can be easily improved with filtering or evolutionary strategies.

---

### Official Review · Reviewer_kwjH · 2025-11-09

**Soundness:** 1
**Presentation:** 1
**Contribution:** 1
**Rating:** 0
**Confidence:** 5

**Summary:**

This paper contains a github handle likely pointing to one of the authors. It violates ICLR's anonimity policy and should be desk rejected.

**Strengths:**

See above.

**Weaknesses:**

See above.

**Questions:**

See above.

---

### Note · Authors · 2025-12-01

I have read and agree with the venue's withdrawal policy on behalf of myself and my co-authors.